# Effects of Parental Migration on Dental Caries of Six- to Eight-Year-Old Children Using Structural Equation Modeling

**DOI:** 10.3390/ijerph192013470

**Published:** 2022-10-18

**Authors:** Sichen Liu, Virasakdi Chongsuvivatwong, Shinan Zhang, Angkana Thearmontree

**Affiliations:** 1Improvement of Oral Health Care Research Unit, Community Dentistry Division, Department of Preventive Dentistry, Faculty of Dentistry, Prince of Songkla University, Songkhla 90110, Thailand; 2Department of Epidemiology, Faculty of Medicine, Prince of Songkla University, Songkhla 90110, Thailand; 3Department of Dental Public Health, School of Stomatology, Kunming Medical University, Kunming 650000, China

**Keywords:** family separation, left-behind children, dental decay, oral hygiene, toothbrushing, dietary sugars, latent class analysis, structural equation modeling

## Abstract

This cross-sectional study aimed to document the relationship between dental caries, oral health behaviors, and the duration of parental migration in rural Yunnan, China, from September to December 2020. Seven rural primary schools with high parental migration were studied. The oral health status of 500 six- to eight-year-old students was assessed using clinical examination and caregivers’ interviews. A total of 51.8% of the children had at least one parent absent for at least 6 months (left-behind children). Among those children with parental migration <6 months, 40.0% consumed sugar twice or more daily and 82.8% of those with parental migration from 6 to 12 months brushed once a day or less. The percentage of daily sugar consumption twice or more and brushing once or less among those without parental migration were 36.0% and 68.6%. Prevalence of caries in permanent teeth (DMFT) in children without parental migration and those whose parental migration <6 months, 6 to <12 months, and ≤12 months were 30.9%, 20.0%, 28.7% and 19.8%, respectively. Out of several other causal pathways between parental migration and dental caries, our structural equation model delineated that sugar consumption is the important mediator variable. Special education programs may be needed to educate caregivers on sugar consumption for the left-behind children in rural areas.

## 1. Introduction

With the emerging economy, China is leading the world in urban migration. According to the latest population census of China, in 2020, the number of the population who have been away from their household registration place for over 6 months reached 492.76 million [1]. In the trend of urban migration, as the result of parents migrating, children were often left behind in the care of other family members or caregivers. These children are often called “left-behind children (LBC)” [2].

The number of children who experienced parental migration increased from 8.05% (30.09 million) in 2000 to 25% (68.87 million) in 2015 [3]. In Thailand, about 24% of children (equivalent to over 3 million) are left behind with parents migrating to urban areas [4]. In the Philippines, about 27% of children (approximately 9 million) have at least one parent living abroad [2].

A systematic review in 2018 on the impacts of parental migration on the health of children whose mean duration of parental migration was less than 6 months found that those children had higher risks of depression, anxiety, suicidal ideation, conduct disorder, substance use, wasting and stunting compared to children with no parental migration [5]. Several studies confirmed the results [6,7,8]. However, the oral health of children with parental migration has been largely overlooked in research and public policy.

Rural children are known to have poor oral health. According to a meta-analysis of dental caries in China from 1987 to 2013, the pooled prevalence of early childhood caries (ECC) for rural children was 63.5%, whereas 59.5% of urban children had ECC [9]. Specifically, the prevalence of ECC in children aged 5 years in rural and urban areas was 68.2% and 63.3%, respectively [9]. Furthermore, in 2018, the national oral health survey reported that 73.4% of rural children aged 5 years had dental caries, slightly more than peer urban children (70.4%) [10]. Overall, a high prevalence of dental caries and poor oral health behaviours are common among rural children.

Although some reports have described the oral health problems among LBC, they did not analyze the pathways of causation. One study found significant differences between children whose parents had migrated for more than six months and those whose parents had migrated for less than six months in the frequency of snacking [11]. Conversely, two other studies found no significant differences [12,13]. None of these three studies was linked to the final outcome: dental caries. On the other hand, a significantly higher prevalence of dental caries was found between those two groups [14,15,16]. None of these three studies analyzed the role of behavioral intermediate variables in the analysis.

Since dental caries is a multifactorial disease, a wide range of factors, such as the family, affected the individual over time to influence the development and severity of dental caries [17]. Therefore, it is important to apply an appropriate statistical method to cope with the relationship between family, individual, and dental caries. Structural equation modeling (SEM) has often been used to evaluate the multilevel factors and their interrelationship. Three studies on oral health in China used SEM to test the association between socioeconomic status (SES), parents and children’s oral health behaviour and the number of primary teeth caries (dmft) in children [18,19,20]. None of them had parental migration in the model.

With the aforementioned knowledge gap, this study aimed to document the relationship between rural childhood dental caries, oral health behaviours and duration of urban parental migration.

## 2. Materials and Methods

### 2.1. Study Design and Setting

A cross-sectional survey was conducted in Qujing city, Yunnan province of China, from September to December 2020. Appendix A shows the map of the research areas. Qujing, a city with a high number of rural LBC in Yunnan, had reported 163,453 LBC in school in 2017 [21]. It has nine counties, among which three adjacent counties were chosen—Zhanyi county, Huize county and Xuanwei county—because of their similar economic background. The respective 2021 per capita disposable income of rural residents was, respectively, USD 2381 [22], USD 1757 [23], and USD 2248 [24], not very different from the average in Yunnan (USD 2028 [25]). The proportion of rural residents in 2020 in these countries was 51.90% [22], 61.10% [23] and 41.85% [24], respectively.

Dental care fees were out-of-pocket for Chinese children generally [26]. The dentist- to-population ratio of Qujing was 1:24,809 [27]. Among the health care in a rural school, there were one or two school doctors for general health, but none for dentistry. Therefore, rural school children have poor access to health care, especially dental health. In addition, there was a national oral health comprehensive intervention program for Chinese children in 2015 [28]. However, this program only covered some counties in China and did not include our study area. There was no other ongoing public dental health program.

The STROBE guidelines were used to ensure the reporting of this observational study.

### 2.2. Sampling Technique

There were 806 primary schools in the three study counties [29]. For logistic reasons, we first chose 53 schools where the one-way travelling time to the county hospital was less than one hour. Of these 53 schools, we chose those with the highest number of LBC that fulfilled the required sample size. Finally, we had 7 schools, 259 LBC, and 241 non-left-behind children (NLBC), with a total of 500 students.

### 2.3. Sample Sizes

The calculation formula for sample size was as follows: $$n={\left[\sqrt{\bar{p}\bar{q}\left(1+\frac{1}{k}\right)}{\;z}_{1-\alpha /2}+\sqrt{p1q1+\frac{p2q2}{k}\;}{\;z}_{1-\beta }\right]}^{2}/\Delta^{2}$$
$$q1=1-p1,\;\bar{p}=\left(p1+kp2\right)/\left(1+k\right),\;∆=|p1-p2|$$
$$q2=1-p2,\;\bar{q}\;=1-\bar{p},\;\;\;\;\;\;\;\;\;\;\;\;k=1$$

The prevalence of dental caries among children whose parents migrated more than six months (*p*1 = 48.7%) and less than six months (*p*2 = 35.12%) was used [15], owing to no previous data on the prevalence of dental caries in four groups of parental migration duration. The sample size required in the current study was 233 per group, with α = 0.05, 1 − β = 0.80, with an expected non-response rate of 5%. In SEM, a minimum sample size regarding the ratio of the number of cases to the number of model parameters would be 20:1 [30]. A minimum sample size would be 300 with 15 parameters, which was based on the present study’s path analysis model (detail available in Appendix A). Hence, a total of 466 children meets the requirement of the present study’s objective.

### 2.4. Data Collection

Oral health examiners were postgraduate students majoring in dental public health, and research assistants were undergraduate students of oral health at the School of Stomatology, Kunming Medical University.

The oral examination was first performed for children in school. Two trained and calibrated dentists conducted the clinical oral health status assessment with plane dental mirrors in a big classroom. Based on the WHO’s standard for assessment of oral health status [31], the study recorded separately early enamel caries and dental caries. The two were combined as decayed teeth. The caries experiences of deciduous dentition were assessed using the ‘dft’ index, which involves decayed teeth (dt) and filled teeth (ft). The number of missing teeth was not analyzed because they might not be missing due to tooth decay. The caries experiences of permanent dentition were assessed using the ‘DMFT’ index, which involves decayed teeth (DT), missing teeth (MT) and filled teeth (FT). The intraexaminer reproducibility among the three locations was 0.86–0.90. The interexaminer reproducibility among the three locations was 0.83–0.89, indicating their good consistency [31]. Assistants were also trained to record the oral examination results using an oral examination form.

A questionnaire for caregivers was prepared, modified from that used in the National Oral Epidemiological Survey in China [10]. Collected information included SES, children’s oral health behaviours and caregivers’ oral health knowledge and awareness. A pilot study was carried out in two rural schools. Caregivers who could read completed the questionnaire by themselves. Those who could not read were directly interviewed, mostly individually.

### 2.5. Variables

Degrees of caries in permanent and primary teeth were the primary outcome variables. The proximal independent variables included sugar consumption, fluoride toothpaste, tooth brushing and dental attendance. They were considered to mediate preceding causes, including SES, migration, oral health knowledge and awareness. Note that parental migration is the latent variable of two observed variables: parental presence and parental migration duration. Sugar consumption is the latent variable for two observation variables: dessert and sugared drinks. (Details available in the Appendix A).

### 2.6. Statistical Analysis

#### 2.6.1. Data Management

Data entry and validation were performed using EpiData 3.0. R software (Version 4.2.1, R Foundation for Statistical Computing, Vienna, Austria) was used to clean and analyze data.

#### 2.6.2. Descriptive and Univariate Statistics

A chi-squared test was used to compare the oral health behaviour across different parental migration durations. A chi-squared test for trend in proportions was performed to test a linear trend between the prevalence of caries and parental migration durations. Linear regression was used to test for a linear trend in the mean caries experience and parental migration duration.

#### 2.6.3. Structural Equation Modeling (SEM)

Caries both in permanent and in primary teeth were used as the final dependent variables, which are correlated. The prevalence of primary teeth caries was not used in SEM analysis because it was very high. Three main steps of SEM were performed. Firstly, confirmatory factor analysis (CFA) was done to test whether the observed variables act well loading to the latent variables. In CFA, factor loadings of the observed variables loaded by the latent variables were computed; Cronbach’s Alpha coefficient was used to check internal consistency and average variance extracted (AVE) was used to measure the convergent validity. Secondly, the weighted least squares mean and variance adjusted (WLSMV) estimator was used to test the coefficient of path analysis. Thirdly, the model fit was tested using various indexes. The ‘Lavaan’ packages in R were used for CFA/SEM.

### 2.7. Ethics Statement

All caregivers of children signed informed consent forms obtained before the survey. The research study was approved by the Research Ethics Committee (REC), Faculty of Dentistry, Prince of Songkla University (EC6210-038).

## 3. Results

Our study subjects were a mixture of children with different parent migration durations; 241 were children whose parental migration was less than 6 months, and 259 were those with parental migration of six months or more. Further division of parental migration is shown in Table 1. The most common caregivers of LBC were mother and grandparents. Caregivers of LBC had significantly older age and lower education level and fewer white-collar occupations.

Table 2 describes the relationship between the oral health behaviours of rural children and the duration of parental migration. Compared with children without parental migration (36.0%), children with parental migration more commonly consumed sugary foods at least twice a day. Notably, a significantly higher proportion of children with parental migration for 6 to 12 months brushed their teeth less than twice a day (82.8%) (*p* = 0.02). For other behaviours, most children used fluoride toothpaste and had dental attendance in the previous 12 months. More than half of the caregivers had appropriate oral health awareness and knowledge, whether or not the parents migrated. None of these variables was significantly associated with the duration of parental migration.

Table 3 summarizes the prevalence and degrees of dental caries with parental migration duration breakdown. All numbers indicate that the subjects had a very high prevalence and high intensity of dental caries. Only caries in permeant teeth were significantly different among the subgroups.

In the SEM analysis, 427 of all 500 participants were utilized to estimate the model, and 73 were omitted owing to missing data. Appendix A shows that from CFA, four latent variables were constructed. All four latent variables loaded well to the corresponding observed variables. The minor exception was socioeconomic status. It had a relatively small load on education and economics and a related small value of Cronbach alpha (0.43). However, the model was acceptable (acceptable level > 0.40). With these minor CFA fitting problems, the latent variables went further into the path analysis of SEM quite well.

Figure 1 summarizes the SEM with significant coefficients by solid arrows. Parental migration was influenced by SES with a negative association (β = −0.35, *p* < 0.001). The low SES families were more likely to migrate and leave children. Regardless of migration status, SES also directly affected oral health knowledge (β = 0.31, *p* < 0.001) and awareness (β = 0.31, *p* < 0.001), fluoride toothpaste use (β = 0.22, *p* < 0.05) and sugar consumption (β = 0.59, *p* < 0.001). Parental migration, the main hypothesis of our study, directly influenced sugar consumption (β = 0.30, *p* < 0.01) but none of the other behaviours. Sugar consumption was the most important (highest coefficient) determinant of permanent teeth caries (β = 0.20, *p* < 0.01), which was less influenced by other nonsignificant oral health behaviours. Finally, dental attendance, which was not significantly determined by other preceding factors, significantly influenced the number of primary teeth caries (β = 0.16, *p* < 0.01).

The goodness of fit of the model is summarized in Table 4. The chi-square statistic for the model was 83.07 with 65 degrees of freedom (*p* = 0.07). In terms of goodness-of-fit indices, RMSEA was 0.03, SRMR was 0.04, and GFI was 0.99, all suggesting an acceptable fit. Overall, when considering all indices, the model performed relatively well.

## 4. Discussion

This study was conducted in rural China with a very high prevalence of LBC, where parents had left for over a year, and elderly caregivers were at low socioeconomic status. With poor oral health behaviours, especially sugar consumption, the children had a very high prevalence and intensity of both primary and permanent teeth caries. Parental migration was shown to be a significant determinant of permanent teeth caries through sugar consumption. As expected, low SES led to different sorts of poor oral behaviours. Finally, dental attendance was associated with primary teeth caries.

The present study showed a high prevalence of dental caries in all parental migration groups. The prevalence of dental caries in rural school children in China was reported to be consistently high [10,32]. The United Nations has included health and well-being as a part of the Sustainable Development Goals (SDGs). However, the specified goal on oral health including the prevalence of dental caries was not mentioned in the SDGs [33]. Caries prevalence is high in China and worldwide [34]. There is a need to promote oral health as the part of health and SDGs and also accelerate oral health development programs in China and other countries.

Worldwide, it has been demonstrated that poor oral health is associated with low SES. China has recently made good progress in economic development, but the oral health status indicates that rural areas were lagging behind in economic growth. This may indicate rural–urban development gaps, which is also elaborated by the high migration rate of the parents in this study.

SES, a well-known factor affecting behaviors, played an important confounding role in our results. Low SES was associated with parental migration (β = −0.35), but it was positively associated with sugar consumption and fluoride toothpaste use. This confounding effect canceled out the crude association between parental migration and their behaviors (Table 2). Using SEM, the independent effect of parental migration was demonstrated in our analysis. Its effect on permanent dental caries was mediated through sugar consumption. There are possible explanations for this relationship. The absence of parents may mean children were taken care of by grandparents with poor parenting competence [35,36]. However, our SEM model had shown that caregivers’ oral health knowledge and awareness had relatively little effect on sugar consumption. The other studies had reported that LBC had emotional problems, which could lead to cravings for sugar [37,38].

Permanent teeth caries is a serious problem, as it directly affects the mastication function. As many as 30.9% of our subjects without parental migration had this problem. In this study, the only confirmed risk factor was sugar consumption. Previous studies reported that SES and dietary behaviours are directly associated with the number of primary teeth caries (dmft) [19,39]. Our difference from them may be due to the fact that we broke down the oral health behaviours, whereas the other two studies did not [18,20]. None of the other studies included parental migration in the model [18,19,20,39].

Toothbrushing frequency and fluoride toothpaste use in this study had no potential effect on dental caries. Perhaps the toothbrushing process was not well practised by the children owing to poor oral health education. Improper use of fluoride toothpaste such as short duration of brushing and excessive post-brushing water rinsing can lead to inadequate protection by fluoride toothpaste [40].

Only primary tooth caries, the most common problem, was significantly related to dental attendance in this study. On the one hand, none of the other oral health behaviours had any significant effect. It could be explained by the same above-mentioned reasons for permanent teeth caries. On the other hand, dental attendance may have a reverse causal relationship with primary teeth caries. Those with more serious primary teeth caries were more likely to seek dental care services.

Finally, with this complexity of the causal pathway, LBC and NLBC were remarkably different in the SES backgrounds. Crude analysis showed that there was only one difference in behaviors, namely, toothbrushing. In addition, the dental caries was different only in the permanent teeth. These crude results were distorted by the confounding effect of SES. Finally, most behaviors were not confirmed by SEM to be important intermediate factors, except for sugary food consumption. The results imply that further study on the effect of parental migration on children’s health outcomes should consider similar complex causal pathways.

There was a limitation in the present study. Because this study did not use a random sample of the total population, the prevalence of caries is confined to the area where parental migration was common. The study design was cross-sectional, and any interpretation of causation must be interpreted with caution.

## 5. Conclusions

Reaffirmation of poor oral health status in this rural area indicates the need to improve the oral health system. Parental migration had an independent effect on oral health behaviour (sugar consumption). Thus, this group needs special attention for the oral health program.

## Figures and Tables

**Figure 1 ijerph-19-13470-f001:**
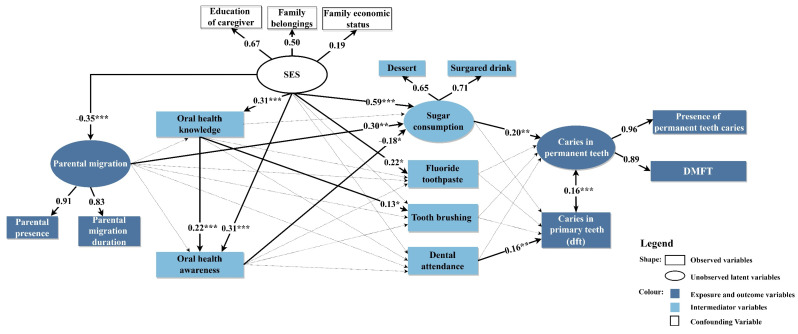
Structural equation model of the relationship between parental migration and dental caries in children. Note: The solid lines indicate significant relationships with the number on each line showing standardized path coefficients; the significant level for path coefficients was set at * *p* < 0.05, ** *p* < 0.01 and *** *p* < 0.001; The dotted lines indicate insignificant relationships.

**Table 1 ijerph-19-13470-t001:** Sociodemographic characteristics of participants, *n* (%).

Characteristic		LBC	NLBC	*p*
Total		259 (51.8)	241 (48.2)	
Parental migration durations				<0.001
	No	-	191 (79.3)	
	<6 months	-	50 (20.7)	
	6–<12 months	87 (33.6)	-	
	≥12 months	172 (66.4)	-	
Parental presence				<0.001
	Both parents	0 (0.0)	191 (79.3)	
	Mother	121 (46.7)	32 (13.3)	
	Father	18 (6.9)	1 (0.4)	
	Grandparents	97 (37.5)	17 (7.1)	
	Others	23 (8.9)	0 (0.0)	
Age of children				0.40
	6	91 (35.1)	83 (34.4)	
	7	132 (51.0)	114 (47.3)	
	8	36 (13.9)	44 (18.3)	
Sex of children				0.19
	Female	118 (45.6)	124 (51.5)	
	Male	141 (54.4)	117 (48.5)	
Age of caregivers				<0.001
	11–29	31 (12.0)	39 (16.2)	
	30–49	118 (45.6)	162 (67.2)	
	50–85	84 (32.4)	25 (10.4)	
Sex of caregivers				0.79
	Female	194 (74.9)	178 (73.9)	
	Male	65 (25.1)	63 (26.1)	
Education level of caregivers				<0.001
	University	2 (0.8)	10 (4.1)	
	High school	19 (7.3)	40 (16.6)	
	Junior high school	92 (35.5)	85 (35.3)	
	Primary school	95 (36.7)	84 (34.9)	
	No formal schooling	51 (19.7)	22 (9.1)	
Occupations of caregivers				0.03
	Officials	1 (0.4)	8 (3.3)	
	Business owners	37 (14.3)	49 (20.3)	
	Farmers	95 (36.7)	81 (33.6)	
	Stay-at-home	119 (45.9)	99 (41.1)	
	Others	7 (2.7)	4 (1.7)	

Note: LBC refers to the children with one or both their parents leaving their home or hometown for work for at least 6 months [8].

**Table 2 ijerph-19-13470-t002:** Crude association between oral health behaviours and parental migration durations, *n* (%).

Variables		Parental Migration Durations (Months)				*p*
		No	<6	6–<12	≥12	
		(*n* = 191)	(*n* = 50)	(*n* = 87)	(*n* = 172)	
Frequency of sugar consumption in children						0.44
	Few/never	83 (44.6)	23 (46.0)	34 (40.0)	63 (36.8)	
	Once a day	36 (19.4)	7 (14.0)	23 (27.1)	42 (24.6)	
	Twice or more	67 (36.0)	20 (40.0)	28 (32.9)	66 (38.6)	
Frequency of brushing in children						0.02
	Twice or more	58 (31.4)	19 (38.0)	15 (17.2)	59 (34.5)	
	Less than twice	127 (68.6)	31 (62.0)	72 (82.8)	112 (65.5)	
Fluoride toothpaste used by children						0.05
	Yes	88 (51.5)	17 (38.6)	29 (41.4)	80 (54.4)	
	No *	48 (28.1)	11 (25.0)	15 (21.4)	37 (25.2)	
	Don’t know	35 (20.5)	16 (36.4)	26 (37.1)	30 (20.4)	
Dental attendance of children in the past year						0.27
	Yes **	119 (62.3)	25 (50.0)	47 (54.0)	94 (54.7)	
	No	72 (37.7)	25 (50.0)	40 (46.0)	78 (45.3)	

Notes: *: No, including toothpaste without fluoride and did not use toothpaste. **: Yes, including dental attendance within 6 months, 6 to 12 months, more than 12 months.

**Table 3 ijerph-19-13470-t003:** Test linear trend in children’s dental caries among parental migration durations.

Variables		Parental Migration Durations (Months)				*p*
		No	<6	6–<12	≥12	
		(*n* = 191)	(*n* = 50)	(*n* = 87)	(*n* = 172)	
Prevalence of caries, % *						
	Caries in permeant teeth	30.89	20.00	28.74	19.77	0.03
	Caries in primary teeth	94.24	94.00	95.40	91.86	0.43
	Total caries	94.76	94.00	95.40	91.86	0.31
Number of caries, mean **						
	Caries in permeant teeth (DMFT)	0.68	0.38	0.55	0.39	0.39
	Caries in primary teeth (dft)	6.90	7.24	7.03	6.85	0.95
	Total caries (DMFT + dft)	7.58	7.62	7.59	7.24	0.88

Notes: Caries was detected as both early enamel caries and dentine caries; DMFT: decayed, missing, and filled permanent teeth; dft: decay, filled primary teeth; *: *p*-values by chi-squared test for Trend in Proportion; **: *p*-values of Linear Regression to test a linear trend in the mean caries experience and parental migration durations, which was a continuous variable (by month).

**Table 4 ijerph-19-13470-t004:** Goodness of fit measures of the model.

Fit Index	Recommend Levels	This Model
χ^2^/df	<5.00	1.28
RMSEA	<0.08	0.03
SRMR	<0.08	0.04
GFI	>0.90	0.99

χ^2^/df: the chi-squared fit statistic; RMSEA: root-mean-square error of approximation; SRMR: standardised root mean square residual; GFI: goodness-of-fit statistic.

## Data Availability

Data is available from the corresponding author for reasonable reasons.

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
