# Peer review of "Effects of Parental Migration on Dental Caries of Six- to Eight-Year-Old Children Using Structural Equation Modeling"

_ijerph, 2022, doi:10.3390/ijerph192013470_

Round 1

Reviewer 1 Report

The manuscript developed a structural equation modeling for children to explore the effects of parental migration on dental caries, which is quite interesting and meaningful. The experimental process was recorded in detail, and followed the STROBE Statement Checklist of items well. Some comments have been provided for your attention:

1.      Abstract:

1.1       Describe the locations, and relevant dates, study size.

2.      Sampling Technique:

2.1       Reasons for choosing schools where the one-way travelling time to the county hospital was less than one hour.

2.2       It is recommended to give a description of the total sample size

2.3       This is the first time that the abbreviation of NLBC appears. The full name of NLBC did not appear in the previous text.

3.      Sample Sizes

3.1       It is recommended to show the calculation formula of sample size.

3.2       In this study, parents were divided into 4 groups according to parental migration durations (No/<6m/6-12m/>=12m). Why calculate sample size using the prevalence of dental caries among children whose parental migrated more than six months or less than six months” (two groups).

3.3       What is the function of Appendix Figure 3 and 4? There is no corresponding explanation in the article. Moreover, the values in sample size does not match Appendix Figure 4.

3.4       Reasons for the ratio of LBC to NLBC was 1: 1

4.      Data Collection

4.1       How to ensure that the results of the interview and independent responses are consistent, and whether there is a consistency test.

4.2       According to my experience of the survey, in fact many caregivers are not clear whether toothpaste contains fluoride. Why this option was not considered when setting this question?

5.      Results

5.1       The prevalence should be checked: Caregivers of LBC had significantly more ageing (36.1%). It is suggested to add the prevalence after more low education level.

5.2       Line 182: The prevalence of children who had sugar consumption twice a day without parental migration was 36.0%. I think it should be calculated by 67/(67+20+28+66)*100.

5.3       Line182-183: “sugar consumption twice a day”: Is the description here should be “sugar consumption twice or more than twice a day”?

5.4       The letter “P” in P value should use the uppercase oblique body, it is recommended to check the full text.

Reviewer 2 Report

Comments

It is an interesting topic to document the relationship between rural children in dental caries, oral health behaviours and duration of parental migration, but it is difficult to evaluate the study based on the information that is provided in the current version of the manuscript.

More clarification is needed on study results:

1. In table 1, 2 and 3, Chi-squared test was used to compare more than two groups, when the differences of average levels among several groups are statistically significant, the multiple comparisons should be further carried out. For example, age of caregivers, education level of caregivers and occupations of caregivers in table 1; Frequency of brushing in children in table 2; Caries in permeant teeth in table 3.

2. There were many unreasonable explanations for the results.

1) Lines 272-273: “The independent effect of parental migration was demonstrated in our analysis. Its effect on dental caries was mediated through sugar consumption.” But the results in table 2 showed that no significant difference in the frequency of sugar consumption among the children with parental migration durations, and the results in table 3 showed that no significant difference in the prevalence of caries in primary teeth and the total caries. Moreover, the prevalence of caries in permanent teeth in children with parental migration ≥ 12 months was the highest.

It should provide a more convincing explanation.

2) Lines 277-278: It was cited “The other studies had reported that LBC had emotional problems, which could lead to cavities for sugar”.

Lines 25-26: It was concluded that “Special education programs may be needed to educate caregivers on sugar consumption for the left-behind children in rural areas”.

In the main text, it seems that it focus on the impact of parental migration durations on the children’s dental caries, oral health behaviours, rather than make comparisons between the LBC and Non-LBC. Therefore, the author should further make comparisons between the LBC and Non-LBC.

In a word, the results didn’t support the conclusion. A statistical analysis should be re- conducted.

Round 2

Reviewer 2 Report

The authors have now addressed the concerns I previously had with the manuscript.